# Application of Near-Infrared Spectroscopy to Detect Modification of the Cation Exchange Properties of Soils from European Beech and Silver Fir Forest Stands in Poland

**DOI:** 10.3390/ijerph20032654

**Published:** 2023-02-01

**Authors:** Mateusz Kania, Dawid Kupka, Piotr Gruba

**Affiliations:** Department of Forest Ecology and Silviculture, University of Agriculture in Krakow, Al. 29 Listopada 46, 31-425 Krakow, Poland

**Keywords:** cation exchange capacity, near-infrared spectroscopy, soil organic matter, soil properties

## Abstract

This study investigated changes in the composition of the cation exchange capacity of soil samples caused by the acid leaching of soil cations under laboratory conditions. Furthermore, near-infrared (NIR) spectroscopy was used to evaluate the properties of forest soils. The potential influence of the species composition of stands (beech and fir) was also investigated. Eighty soil samples from the topsoil of plots located in central Poland were analyzed. Soil samples were leached 0 (non-leached), 5, 10, and 15 times and then analyzed to determine the contents of cations (Al^3+^, Ca^2+^, K^+^, and Mg^2+^), the total carbon content, and the pH. From NIR spectra obtained by scanning 54 samples and measurement results for soil sample properties, a calibration model was developed. The model was validated using 26 independent samples. The results showed that acid leaching decreased the pH of soil solutions and the carbon content. The amounts of Al^3+^, Ca^2+^, K^+^, and Mg^2+^ decreased with an increasing number of leaching treatments, but most leaching had occurred after five treatments. Data analysis showed that leaching with hydrochloric acid depleted alkaline cations and Al^3+^ in the soil, which reduced the stability of organic matter, causing its release. Modification of ion exchange properties is observable based on the analysis of the NIR spectra. Good calibration results were achieved for all tested parameters (R^2^C ≥ 0.89). The best validation results were obtained for Al^3+^ and C contents under fir stands, and for the pH and Al^3+^ content of soils under beech stands (R^2^V > 0.8). However, the differences between the measured and estimated mean values of the investigated soil were relatively small (no significant difference, *p* > 0.05). The species composition of stands (beech and fir) had no impact on the developed mathematical models. Soil assessment using NIR spectroscopy allowed calibration models to be obtained, which were successfully used to calculate soil properties at a much lower cost and in a much shorter time compared with other laboratory methods. The results of the paper affirmed that using a relatively small number of samples (3–4) to calculate an average of soil content properties provided satisfactory results.

## 1. Introduction

Forest soil acidification is a common process caused by many factors, including atmospheric precipitation that reaches the soil surface. Owing to changes in rainwater chemistry that occur during runoff on trunks, by passage through tree crowns, and on the forest floor, water can acidify soil. This effect is particularly intensive in soil containing a thick layer of acidic organic horizon. Therefore, the species composition of undergrowing vegetation affects the soil acidity in forest ecosystems [1,2]. The strong influence of water and dissolved chemical compounds on the chemical properties of soils can be clearly observed in the vicinity of a tree’s trunk and becomes weaker further from the tree [3,4]. Atmospheric precipitation running down the trunk or passing through the canopy becomes enriched or depleted in alkaline or acidic ions [5,6]. Coniferous stands are believed to be more efficient at capturing aerosols from the air than deciduous stands [7]. Therefore, acid deposition and changes in the forest species composition have the potential to alter the chemical properties of soils.

These interactions can be compared to weak acid leaching, which causes changes in the cation exchange capacity (CEC). Leaching has been observed to remove aluminum (Al^3+^) and base cations, such as calcium (Ca^2+^), potassium (K^+^), and magnesium (Mg^2+^), which are replaced with hydrogen (H^+^), as reported, for example, by Berggren et al. [8]. Changes in the proportions of cations in the CEC might increase or reduce the stability of organic matter [8,9,10]. Furthermore, changes in the composition of the soil CEC and the stability of soil organic matter (SOM) depend on the intrinsic properties of SOM inherited from dominant tree species.

To date, changes in the cation composition of the soil CEC and its effect on the stability of SOM have usually been investigated using chemometric methods. However, insufficient attempts have been made to analyze these changes using near-infrared (NIR) spectroscopy.

NIR spectroscopy is a dynamically developing technique based on the interpretation of oscillation–rotation spectra of a given substance [11,12,13]. NIR spectroscopy is a rapid and inexpensive method, and it provides an alternative to expensive and time-consuming laboratory measurements. NIR spectroscopy has been applied to estimate many soil attributes, such as soil organic matter, carbon content, nitrogen content, pH, texture, and moisture [13,14,15]. The low cost of measurement results from the simple soil-sample preparation required for NIR analysis, while easy operation means that analysis can be performed by personnel after initial training. The NIR technique enables the estimation of soil properties in couple of seconds with use of a single spectrum [16,17,18]. Although NIR spectroscopy has rarely been applied to investigate forest soils, this technique could facilitate the estimation of forest soil properties [19,20].

Therefore, it is valuable to try to develop alternative analytical methods that will replace laboratory techniques [15,18,21]. Few attempts have been made to determine CEC to date. The good validation results in multiple papers suggest that the NIR method can be used to determine the chemical and physical properties of soils [22,23,24].

The results obtained using NIR spectroscopy are only a model estimate of the real values and are prone to error. Therefore, a large number of samples is beneficial, allowing an average value to be obtained that is statistically similar to the values obtained by laboratory measurements. In particular, little is known about whether the species composition of the forest stand impact the developed models of forest soils [15,25].

This study aimed to investigate changes in the composition of the CEC of soil samples caused by acid leaching of soil cations under laboratory conditions. To date, the literature has not provided a clear indication of whether the quality of organic matter, related to the species composition of the stand, has any impact on cation leaching and organic matter stability. Therefore, to determine the possible effects of tree species, samples originating from two types of forest stand, European beech and silver fir, were tested. Furthermore, this study aimed to use NIR spectroscopy to evaluate changes in the pH, CEC composition, and carbon content of soil. To achieve this goal, the soil properties obtained by NIR spectroscopy were compared with those determined using chemometric methods.

## 2. Materials and Methods

### 2.1. Study Site

Twenty research plots were established in the Holy Cross Mountains, located in central Poland. Ten plots covered with European beech (*Fagus sylvatica* L.) stands and ten plots covered by silver fir (*Abies alba* Mill.) stands were established. For all plots, homogeneous areas were selected. The elevation of the study area ranged between 390 to 412 m above sea level. The soil was classified as Dystric Cambisol, developed from Triassic sandstones and claystones [26,27]. Each sample was collected from mineral topsoil (A), located at the center of the plots (average depth, 12 cm). A total of 20 soil samples were collected.

### 2.2. Laboratory Analysis

Each soil sample (20 g) was added to an individual 50 mL plastic tube. HCl solution (40 mL, 1 × 10^−3^ M) was added to each tube. After mixing the suspension, the tubes were centrifuged at 3000 rpm for 20 min. The clear supernatants were then poured into plastic bottles. Soil remaining in the test tubes was mixed with another 30 mL of the HCl solution to obtain another portion of supernatant from the same soil sample. This procedure was repeated 5, 10, or 15 times to obtain the first, second, and third series of samples, respectively.

The soil samples (leached 0 (non-leached), 5, 10, or 15 times) were analyzed to determine the concentrations of soil cations (Al^3+^, Ca^2+^, K^+^, and Mg^2+^), the total carbon (C) content, and the pH. First, the soil samples were ground by a ball mill (Fritsch, Idar-Oberstein, Germany) to obtain homogeneity. These finely ground samples were then used to measure the total C content using a LECO CNS TruMac analyzer (Leco, St. Joseph, MI, USA). The soil pH was measured by a potentiometric method using a combined electrode in a soil suspension in distilled water (1:5 mass-to-volume ratio) after equilibration for 24 h [28]. The exchangeable aluminum (Al^3+^) concentration was measured by inductively coupled plasma optical emission spectrometry (ICP-OES; Thermo iCAP 6500 Duo, Thermo Fisher Scientific, Cambridge, UK) after extracting the soil sample (3 g) with CuCl_2_ solution (30 mL, 0.5 M) by shaking for 2 h [29]. Exchangeable calcium (Ca^2+^), potassium (K^+^), and magnesium (Mg^2+^) were extracted with CH_3_COONH_4_ solution (1 M, pH 7). Samples (10 g) were mixed with the extractant (30 mL), shaken for 1 h, and then equilibrated for 24 h. The samples were then filtered and made up to a volume of 100 mL with the extractant solution, and the cation concentration was measured by ICP-OES [30].

### 2.3. NIR Spectroscopy Analysis

NIR spectroscopy was performed using a Fourier transform (FT) NIR spectrometer (Antaris II FT-NIR; Thermo Fisher Scientific, Waltham, MA, USA). In total, 80 soil samples (20 samples × 4 variants (leached 0, 5, 10, and 15 times)) were subjected to scanning for several seconds. Soil samples were placed in a glass tube (height, 5 cm; diameter, 1.9 cm) in the spectrometer for analysis. Spectra were collected in the wavelength range of 1000–2500 nm. The spectrometer did not require any additional adjustment. Spectral analysis was performed using the TQ Analyst 8 software (Thermo Fisher Scientific, Waltham, MA, USA). The spectrum files of 54 soil samples were used to calibrate the models. To verify the accuracy and reliability of the developed models, a validation test was performed on 26 systematically selected spectra (every third file) that had not previously been used to calibrate the models. Following repeated test calibrations, the parameters producing the highest correlation coefficients were selected for the studied soil properties. The full range of spectral lengths was used to calibrate the model, and the partial least squares (PLS) regression model was applied. The derivative format and the type and size of the smoothing filters were selected. The quality of the calibration and validation of soils was expressed by the coefficients of determination of the calibration (R^2^C) and validation (R^2^V). The mean values were calculated on the basis of the properties of soils determined by laboratory methods and compared with the values calculated by the NIR method.).

## 3. Results and Discussion

### 3.1. General Characteristic of Investigated Soils

The tested soils (before leaching) were acidic (Table 1), and the average pH values of fir and beech stands were not significantly different. In contrast, the contents of C, exchangeable aluminum (Al^3+^), and other soil cations (Ca^2+^, K^+^, Mg^2+^) significantly differed between stands. Furthermore, the soil samples collected from beech stands had a significantly higher content of sand (ranging from 29% to 70%, average of 56.74%) than those from fir stands (ranging from 6% to 49%, average of 32.8%). Soil samples from fir stands had the highest content of clay (ranging from 47% to 82%, average of 60.7%). Meanwhile, the silt content was low in soils from both forest stands (average of <7%).

### 3.2. Results of Soil Leaching with Dilute HCl Solution

Similar to acid rain, soil leaching with HCl causes the loss of exchangeable fixed cations, such as Al^3+^, Ca^2+^, K^+^, and Mg^2+^ (Figure 1). As a result of the loss of alkaline cations, the soil pH decreases owing to protonation (saturation with H^+^ ions) of the SOM functional groups. The cation leaching dynamics differ depending on the type of cation and the initial concentration. An increasing number of HCl treatments increased the amount of leached Ca^2+^, K^+^, and Mg^2+^, but the vast majority of these ions were washed out after the first five leaching iterations. In contrast, the Al^3+^ content remained relatively stable. A similar experiment involving soil leaching with HCl was conducted by Berggren et al. [8], who reported that cation leaching was accompanied by the destabilization of organic matter and its leaching from the soil as dissolved organic carbon (DOC).

The stabilization of organic matter by metal cations has been the subject of many publications, for example [9,10,23]. This phenomenon is related to cation exchange in the soil and the ability of cations to enhance the formation of other layers of SOM adsorbed to the soil mineral phase. The different rates of depletion in soils under fir and beech stands were attributed to the different organic matter contents and initial cation contents, but also to the specific properties of SOM derived from these two species. According to the literature, soils under coniferous and deciduous stands have different cation neutralization abilities [22]. As the alkaline cation content is related to the organic matter content, soils under coniferous stands (like fir) contain more basic cations and aluminum than those under deciduous stands (like beech). Furthermore, soils under beech stands had a lower relative exchangeable capacity (CEC/C) and lower acidity than soils under coniferous stands (Figure 2) [23]. We observed that the leaching of significant amounts of Al^3+^ from acidic soils contributed to the depletion of a certain pool of aluminum associated with SOM, leading to increased leaching of soil DOC. A similar observation was reported by Berggren et al. [8].

However, this research indicated that soil under fir stands has a greater acidification resistance than that under beech stands, not only owing to the increased buffer properties, but also owing to the specific properties of organic matter from fir stands.

### 3.3. NIR Analysis

#### 3.3.1. General Characteristic of Soil Spectra

Figure 3 shows averaged spectra of soils under fir and beech forest stands after 0, 5, 10, and 15 leaching treatments with HCl solution (1 × 10^−3^ M). All spectra showed characteristic peaks at 7200, 5200, and 4500 cm^−1^. However, the averaged spectra for each soil group were slightly different. Notably, the absorbance value decreased with an increasing number of leaching treatments for soils from both fir and beech stands. The largest differences in absorbance occurred between the spectra of the soils after 5 and 10 leaching treatments. Furthermore, the spectra of soils under beech stands had higher absorbance values (56–65%) compared with the spectra of soils under fir stands (53–65%).

Many attempts have been made to use NIR spectroscopy to predict soil attributes [13]. According to Chang et al. [14], NIR spectroscopy can be used to predict soil properties, and this technique has been used to analyze carbon content and other similar soil properties [14,24]. To date, few attempts have been made to determine the CEC. Analysis of the NIR spectra indicated that modification of the ion exchange properties was observed.

#### 3.3.2. Calibration Results

The calibration results are shown as scatter plots in Figure 4 and Figure 5. Good calibration results, as expressed by the coefficient R^2^C, were obtained for all tested soil parameters. The lowest R^2^C value was obtained for the K^+^ calibration of soils under beech stands. The validation results expressed by the coefficient R^2^V are shown in Table 2. The highest R^2^C values were obtained for the Al^3+^ content and pH of soils under beech stands, and for the Al^3+^ and C contents of soils under fir stands (R^2^V > 0.8). Relatively good results were obtained for the Mg^2+^ and C contents of soils under beech stands, and for the Mg^2+^ content and pH of soils under fir stands. The lowest quality results were obtained for the Ca^2+^ and K^+^ contents of soils under both stands (R^2^V < 0.5) (Table 2).

#### 3.3.3. Validation Results

The discrepancies in the measured and estimated mean values of the investigated soil properties were relatively small. The largest differences between the mean values were observed for the Ca^2+^ and K^+^ contents of soils under beech and fir stands. However, these differences were not statistically significant (Table 3 and Table 4). These results suggested that NIR spectroscopy can be successfully applied to the analysis of soil CEC and its modification. Using a relatively small number of samples (3–4) to calculate an average provided satisfactory results. However, the application of a single-sample prediction method is not recommended. The main disadvantages of laboratory analyses are the high prices of necessary chemical reagents and technical gases, and their labor-intensive and time-consuming nature. Furthermore, an adequately equipped laboratory and trained personnel are required to perform such analyses. Therefore, attempts to develop alternative analytical methods to replace laboratory techniques are valuable [15,18,21]. The good validation results, especially for C and Al^3+^ contents and pH, were consistent with previous reports that suggested that the NIR method can be used to determine the chemical and physical properties of soils [31,32,33].

The species composition of stands influences the chemical and physical properties of soil [34]. The models developed in this study seem to be universal, irrespective of the forest species. This was confirmed by the poor results for Ca^2+^ and K^+^ validation compared with other soil parameters analyzed under both beech and fir stands. Relatively good validation results were obtained for cations, C, and pH (R^2^V > 0.70), except for the average result of the validation for Mg^2+^ in soils under fir stands. The beech and fir stands showed no influence on the results of the NIR method. Therefore, the species composition of stands does not seem to affect the soil models [19].

## 4. Conclusions

Acid leaching of soils leads to a decrease in the pH of soil solutions and a slight decrease in the C content. Most cations, particularly Ca and Mg, were leached after the first five leaching treatments, while Al and DOC were released gradually. Data analysis showed that leaching with HCl caused the depletion of alkaline and aluminum cations in the soil, which reduced the stability of organic matter, causing its release. The application of NIR spectroscopy to soil assessment allowed calibration models to obtained, which were successfully used to calculate soil properties, such as cation exchange properties, at a much lower cost and in a much shorter time compared with previous laboratory methods. This could reduce the total cost of soil work. The species composition of stands (beech and fir) had no potential impact on the developed mathematical models, which seem to be universal.

## Figures and Tables

**Figure 1 ijerph-20-02654-f001:**
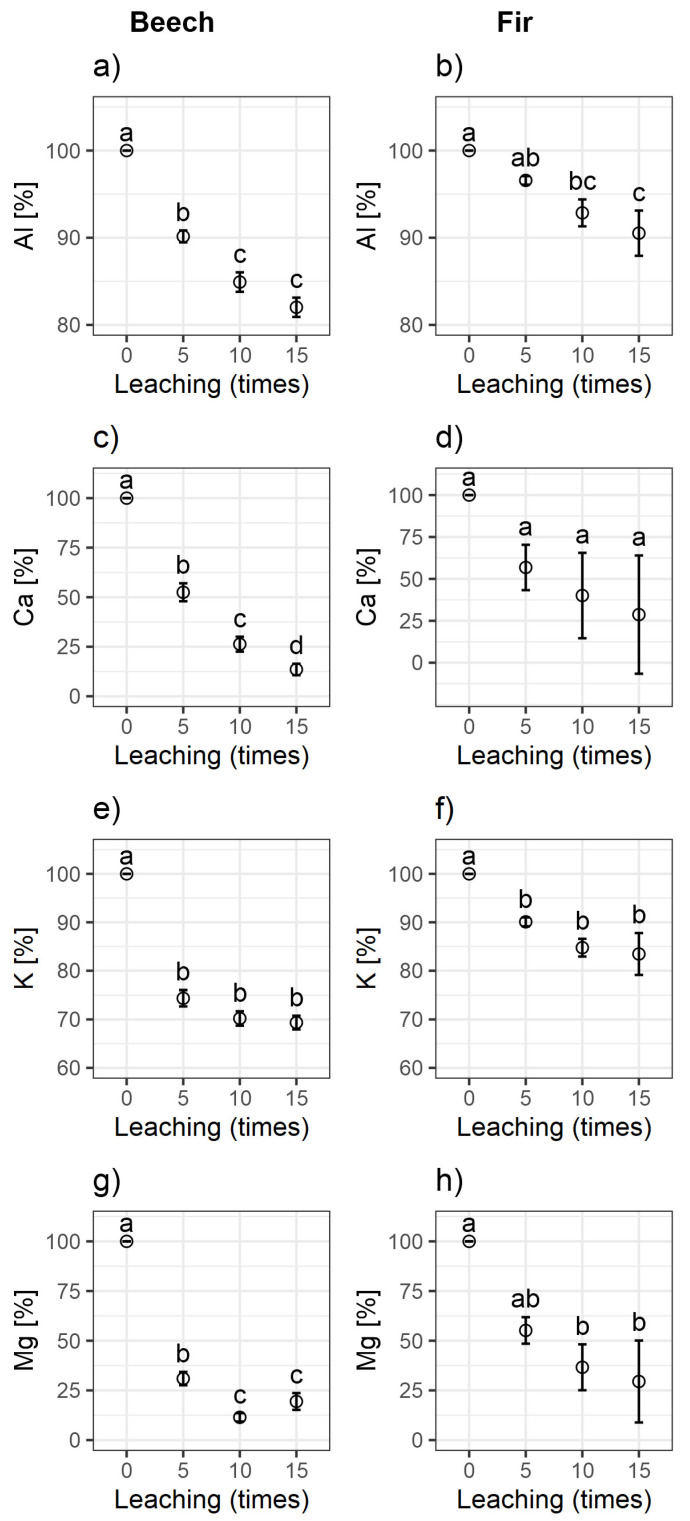
Comparison of percentage changes in the contents of soil cations (**a**) Al^3+^, (**c**) Ca^2+^, (**e**) K^+^, and (**g**) Mg^2+^ in soils under beech and (**b**) Al^3+^, (**d**) Ca^2+^, (**f**) K^+^ and (**h**) Mg^2+^ in soils under fir stands as a result of leaching (0 (non-leached), 5, 10, and 15 times).

**Figure 2 ijerph-20-02654-f002:**
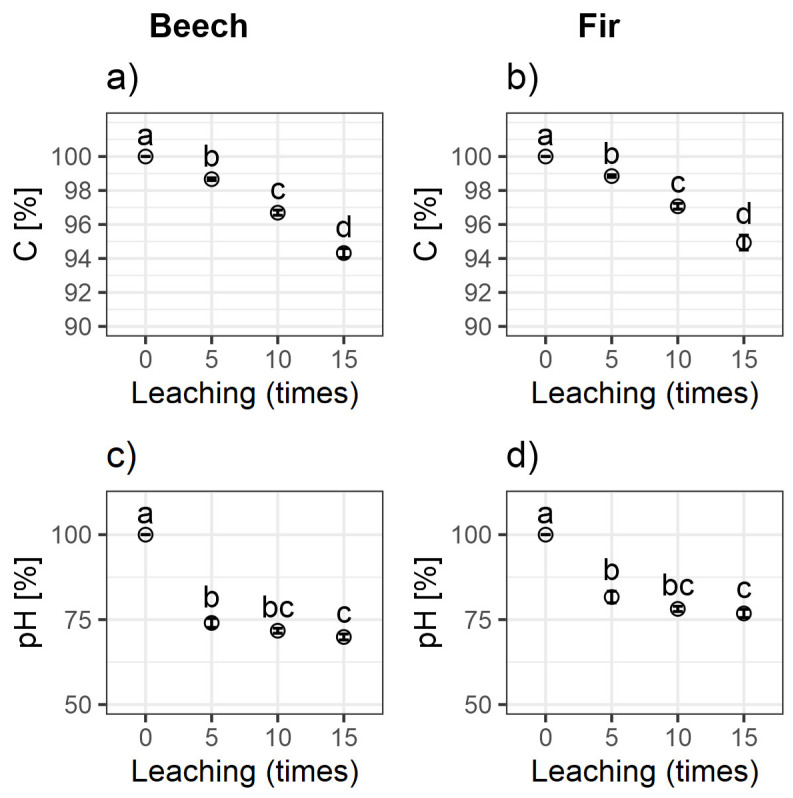
Comparison of percentage changes in the (**a**) carbon content and (**c**) pH of soils under beech stands and the (**b**) carbon content and (**d**) pH of soils under fir stands as a result of leaching (0 (non-leached), 5, 10, and 15 times).

**Figure 3 ijerph-20-02654-f003:**
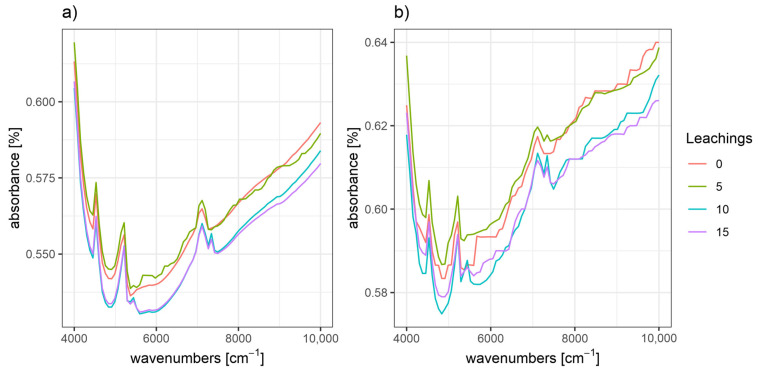
Averaged spectra of soil samples under (**a**) beech and (**b**) fir stands after leaching (0 (non-leached), 5, 10, and 15 times).

**Figure 4 ijerph-20-02654-f004:**
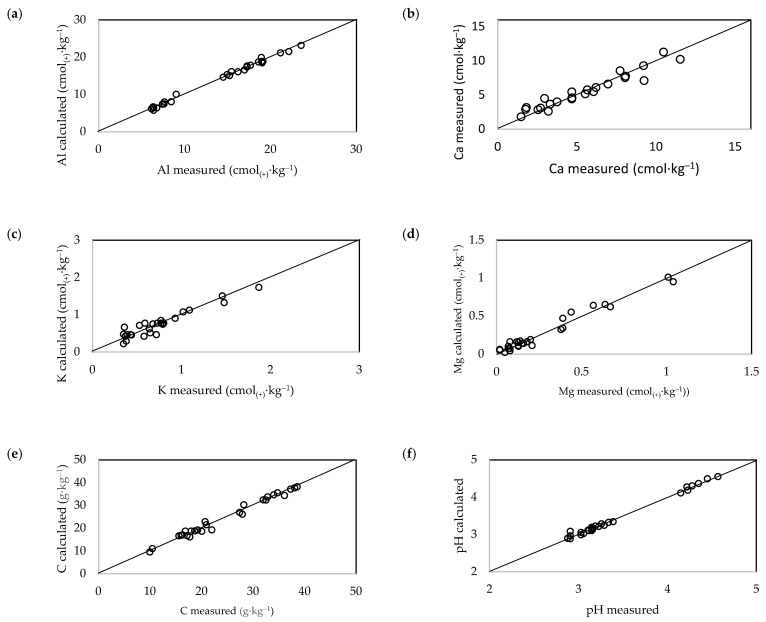
Calibration scatter plots of measured versus predicted values for (**a**) Al^3+^, (**b**) Ca^2+^, (**c**) K^+^, (**d**) Mg^2+^, and (**e**) C contents, and (**f**) pH of soil under beech stands.

**Figure 5 ijerph-20-02654-f005:**
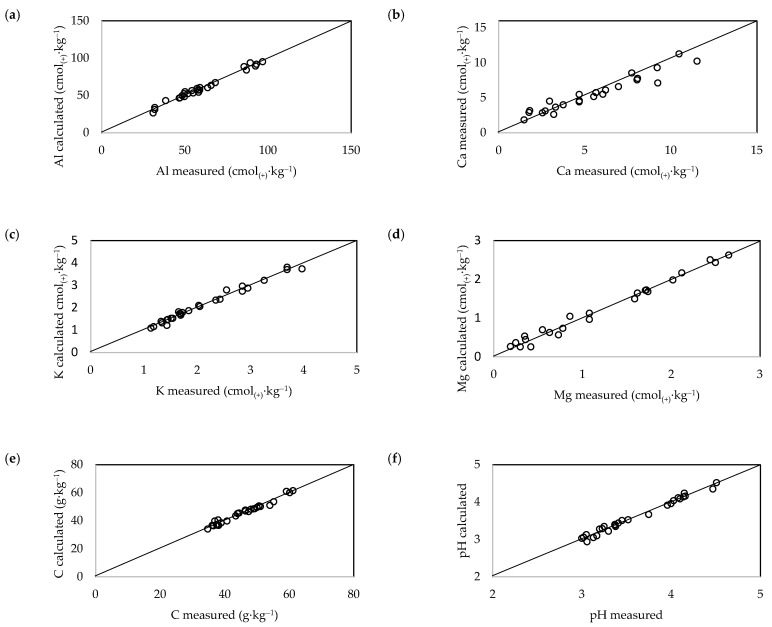
Calibration scatter plots of measured versus predicted values for (**a**) Al^3+,^ (**b**) Ca^2+^, (**c**) K^+^, (**d**) Mg^2+^, and (**e**) C contents, and (**f**) pH of soil under fir stands.

**Table 1 ijerph-20-02654-t001:** Exchangeable cation (Al^3+^, Ca^2+^, K^+^, and Mg^2+^) contents, carbon content, and pH of non-leached soils (average values *±* standard error, *n* = 10) in soil under beech and fir stands.

	Al^3+^	Ca^2+^	K^+^	Mg^2+^	C [g∙kg^−1^]	pH
	[cmol_(+)_∙kg^−1^]					
Beech	15.62 ± 6.69	2.94 ± 1.52	0.91 ± 0.40	0.64 ± 0.24	24.2 ± 9.26	4.35 ± 0.13
Fir	61.62 ± 19.09	6.48 ± 3.20	2.26 ± 0.83	1.70 ± 0.67	46.3 ± 8.14	4.28 ± 0.25

**Table 2 ijerph-20-02654-t002:** Results of calibration and validation (Al^3+^, Ca^2+^, K^+^, Mg^2+^, and C contents, and pH), expressed by R^2^C and R^2^V.

Soil Properties	Al^3+^	Ca^2+^	K^+^	Mg^2+^	C	pH
*Fagus sylvatica* L.						
R^2^C ^a^	0.99	0.97	0.89	0.97	0.99	0.98
R^2^V ^b^	0.87	0.33	0.47	0.72	0.88	0.70
*Abies alba* Mill.						
R^2^C	0.98	0.92	0.98	0.98	0.98	0.97
R^2^C	0.83	0.38	0.47	0.56	0.74	0.84

^a^ R^2^C coefficient of determination for calibration; ^b^ R^2^V, coefficient of determination for validation.

**Table 3 ijerph-20-02654-t003:** NIR spectroscopy results for the validation of soils under beech stands (measured versus calculated values, *p*).

Soil Properties ^a^	Number of Leaching Treatments								
	0 (*n* ^b^ = 3)		5 (*n* = 3)		10 (*n* = 3)		15 (*n* = 4)		
	Mean Measured	Mean Calculated	Mean Measured	Mean Calculated	Mean Measured	Mean Calculated	Mean Measured	Mean Calculated	*p* ^c^
Al^3+^	14.99	15.32	14.47	12.74	12.92	12.83	13.05	12.35	NS
Ca^2+^	2.56	3.38	1.34	1.47	0.61	0.38	0.59	2.02	NS
K^+^	0.81	0.79	0.80	0.58	0.58	0.37	0.61	0.77	NS
Mg^2+^	0.55	0.55	0.18	0.25	0.05	0.10	0.10	0.18	NS
C	17.31	18.89	24.53	20.61	16.71	17.54	27.45	29.91	NS
pH	4.40	4.17	3.23	3.30	3.10	2.95	3.06	3.21	NS

^a^ Al^3+^ [cmol_(+)_∙kg^−1^]; Ca^2+^ [cmol_(+)_∙kg^−1^]; K^+^ [cmol_(+)_∙kg^−1^]; Mg^2+^ [cmol_(+)_∙kg^−1^]; C, total carbon content [g∙kg^−1^]; ^b^ *n*, number of samples; ^c^ *p*, significance level; NS, no significant difference (*p* > 0.05).

**Table 4 ijerph-20-02654-t004:** NIR spectroscopy results for the validation of soils under fir stands (measured versus calculated values, *p*).

Soil Properties ^a^	Number of Leaching Treatments								
	0 (*n* ^b^ = 3)		5 (*n* = 3)		10 (*n* = 3)		15 (*n* = 3)		
	Mean Measured	Mean Calculated	Mean Measured	Mean Calculated	Mean Measured	Mean Calculated	Mean Measured	Mean Calculated	*p* ^c^
Al^3+^	52.33	52.69	59.46	58.65	46.06	52.93	62.59	60.99	NS
Ca^2+^	5.37	5.70	5.06	3.98	2.97	3.71	4.02	4.20	NS
K^+^	1.72	2.01	2.49	1.76	1.38	1.85	1.94	2.16	NS
Mg^2+^	1.43	1.31	1.18	0.92	0.29	0.56	0.83	0.82	NS
C	42.17	44.62	51.26	49.39	40.83	43.90	43.33	44.54	NS
pH	4.48	4.14	3.36	3.50	3.57	3.33	3.20	3.40	NS

^a^ Al^3+^ [cmol_(+)_∙kg^−1^]; Ca^2+^ [cmol_(+)_∙kg^−1^]; K^+^ [cmol_(+)_∙kg^−1^]; Mg^2+^ [cmol_(+)_∙kg^−1^]; C, total carbon content [g∙kg^−1^]; ^b^ *n*, number of samples; ^c^ *p*, significance level; NS, no significant difference (*p* > 0.05).

## Data Availability

The data presented in this study are available on request from the corresponding author.

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
