# Peer review of "Application of Near-Infrared Spectroscopy to Detect Modification of the Cation Exchange Properties of Soils from European Beech and Silver Fir Forest Stands in Poland"

_ijerph, 2023, doi:10.3390/ijerph20032654_

Round 1

Reviewer 1 Report

1.      The exchange times for calcium and potassium and magnesium should be shown because it will affect the contents of these cations. I suggest to cite the references for the methods.

2.      I suggest to show the scan times for the soil samples.

3.      Figure 1 and 2: Please write down the unit for x-axis caption.

Author Response

Reply in the attachment.

Reviewer 2 Report

The subject of the manuscript is consistent with the scope of the Journal. The statistical methods appear to have been applied appropriately and the manuscript is generally clearly written. However, I'm afraid the novelty of the paper is not fully exploited and the authors should considerably improve and strengthen the novelty of the paper. The authors need to add more results and their findings in the abstract. The introduction must be focused on the objectives and methods. 

Author Response

Reply in the attachment.

Reviewer 3 Report

The manuscript investigated changes in the composition of the CEC of soil samples caused by acid leaching of soil cations under laboratory conditions. Furthermore, NIR was used to evaluate the properties of the forest soils. The potential influence of the species composition of stands was also investigated. The research is meaningful and novel. However, after careful reading the manuscript, I’m not convinced the contributions are enough for the publication. The following are some comments for the authors to improve the manuscript.

Major issues:

1. Hyperspectral spectrum usually plays better in 400nm – 1000nm than NIR when analyzing cations. Although NIR also plays good, it seems to be better to use both 400-1000nm as well as the NIR for this study. NIR can be analyzed more than 400-1000nm.

2. Table 2 shows many of the results have low R2v lower than 0.7, especially for Ca2+, K+ and Mg2+. This means the NIR models are not so confident. The authors might be able to improve the modelling, or even try my suggestions in 1 to add 400nm-1000nm.

3. The strongest strength of hyperspectral RS is that is can mapping the whole study areas rather than the only analysis of the samples. Therefore it would be better if this study can be promoted to hyperspectral images if the data are accessible.

4. The study used 54 samples to calibrate the model and used another 26 samples to verify the accuracy among the total 80 samples. However, when the samples are small, ML modelling can use the samples mutually several times and then integrate into the final models.

Minor issues:

1. The gramma of the manuscript should be improved and the description jumped many times. For example, many citations appear at the Results and Discussion part. They should be moved to the Introduction part.

2. In Fig 2 – Mg, the percentage of the Mg increased after the 15 leaching. It seems to be the only case for the whole study. There should be some special analysis.

3. The averaged spectra of soil samples in Fig 3 are in cm-1. It should be better for reading if the authors use nm.

Author Response

Reply in the attachment. 

Round 2

Reviewer 3 Report

The authors have tried meaningful work in predicting CEC using NIR. Although the research can be improved greatly ideally, the authors have done most of the work under the research condition they can access. I therefore reckon if the editor thinks this research is relative with the topics of the journal, this manuscript can be acceppted after minor gramma and text reading revision.